# Vitamin D and Histological Features of Breast Cancer: Preliminary Data from an Observational Retrospective Italian Study

**DOI:** 10.3390/jpm12030465

**Published:** 2022-03-14

**Authors:** Stefano Lello, Anna Capozzi, Lorenzo Scardina, Lucia Ionta, Roberto Sorge, Giovanni Scambia, Gianluca Franceschini

**Affiliations:** 1Department of Woman and Child Health and Public Health, Institute of Obstetrics and Gynecology, Fondazione Policlinico Universitario Agostino Gemelli IRCCS, Università Cattolica del Sacro Cuore, 00168 Rome, Italy; stefano.lello@policlinicogemelli.it (S.L.); anna.capozzi@policlinicogemelli.it (A.C.); giovanni.scambia@policlinicogemelli.it (G.S.); 2Department of Woman and Child Health and Public Health, Division of Breast Surgery, Fondazione Policlinico Universitario Agostino Gemelli IRCCS, Università Cattolica del Sacro Cuore, 00168 Rome, Italy; lucia.ionta@guest.policlinicogemelli.it (L.I.); gianlucafranceschini70@gmail.com (G.F.); 3Laboratory of Biometry, Department of Systems Medicine, University of Rome Tor Vergata, 00187 Rome, Italy; sorge@uniroma2.it

**Keywords:** vitamin D, breast cancer, ductal breast cancer, in situ breast cancer, lobular breast cancer, histology

## Abstract

Background: Vitamin D (vitD) may be involved in different extraskeletal conditions as well as skeletal muscle diseases. It has been hypothesized that, at least in part, a low level of vitD could contribute to facilitating cancer development. Breast cancer (BC) seems to be associated with low levels of vitD. Materials and methods: This was an observational retrospective evaluation of 87 women (mean age: 54 ± 12 years old) who underwent surgery for the treatment of BC. Our main purpose was to correlate the types of BC and the levels of vitD. Results: A positive significant correlation (R > 0.7) was found between non-invasive carcinoma in situ and 25(OH)D levels and age (R = 0.82, *p* < 0.05). A positive, but nonsignificant, correlation was reported between invasive ductal carcinoma and 25(OH)D and age (R = 0.45, *p* > 0.05). A negative but nonsignificant correlation was found between invasive lobular carcinoma and 25(OH)D and age (R = 0.24, *p* > 0.05). Discussion and Conclusions: We did not find a significant relationship between vitD and BC subtypes. Considering the positive significant correlation between vitD levels and age for in situ BC, although preliminary, our results seem to suggest a possible role of vitD in in situ BC. However, these findings need to be confirmed in larger studies.

## 1. Introduction

Breast cancer (BC) is the most common form of female cancer and the second leading cause of death among women worldwide [1]. Many data suggest that several lifestyle and environmental factors—such as a high-fat diet, a lack of physical activity, and chronic alcohol consumption—might play a critical role in the development and risk of recurrence of this cancer [2]. The maintenance of a healthy lifestyle together with regular breast checks, through breast self-examinations, mammography, and/or ultrasonography, represent a cornerstone for primary prevention [3]. Vitamin D (vitD) homeostasis is fundamental for the achievement of bone strength and the prevention of bone and muscle loss [4].

According to the principal international guidelines [5,6], its supplementation is advisable for frail subjects affected by a loss of bone strength and/or hypovitaminosis D. Furthermore, emerging data demonstrate that vitD may produce other important benefits at the extraskeletal level [7]. Hypovitaminosis D is associated with a higher incidence of several cardiovascular, metabolic, autoimmune, endocrine, and neoplastic pathologies [7]. Although large intervention studies about the positive effects of vitD supplementation on these pathologies are limited [8], there is growing interest in the potential involvement of vitD status in the appearance of some of these diseases [7]. 

An interesting meta-analysis by Hossain et al. [9] showed a direct correlation between vitD deficiency and BC, with a relative risk (RR) of 1.91 (95% confidence interval (CI): 1.51–2.41, *p* < 0.001). At the same time, total vitD intake and supplemental vitD intakes had inverse relationships with BC (RR = 0.99, 95% CI: 0.97–1.00, *p* = 0.022, per 100 IU/day; RR = 0.97, 95% CI: 0.95–1.00, *p* = 0.026, respectively). 

Another cohort study [10], evaluating 50,884 women (aged between 35 and 74 years) who had never had BC themselves but had a sister affected by BC, found that high serum 25(OH)D levels and self-reported regular vitD supplementation (≥four times/week) were associated with lower rates of incident BC after menopause over 5 years of follow-up (HR = 0.72 (CI: 0.57–0.93) for high serum 25(OH)D levels, and HR = 0.83 (CI: 0.74–0.93) for regular supplementation).

Our study was an observational retrospective evaluation enrolling 87 women (mean age: 54 ± 12 years old) who underwent surgery for the treatment of BC between December 2019 and March 2020. The objective of this evaluation was to correlate the subtypes of BC with the level of vitD. 

## 2. Materials and Methods

This was an observational retrospective analysis of 87 patients (mean age: 54 ± 12 years old) who had not been supplemented with vitD in the previous 3 months, selected among patients who underwent surgery for the treatment of BC in the Breast Unit Surgery of Fondazione Policlinico Agostino Gemelli IRCCS of Rome. This retrospective analysis included patients without differences in race and/or ethnicity. The exclusion criteria were being a foreign woman and/or being regularly supplemented with vitD analogues in the previous three months before breast surgery and/or being unwilling to be tested for vitD status. 

All the enrolled women were screened for serum 25(OH)D levels during the pre-hospitalization stage. The total serum 25(OH)D levels were measured using an automated Abbott Architect 25(OH)D immunoassay (bias ng/mL% = + 0.4/1.7–4.7 between 20 and 40 ng/mL, according to the Vitamin D External Quality Assurance Scheme).

According to the US Endocrine Society Clinical Practice Guidelines, vitamin D sufficiency was defined as serum levels of 25(OH)D ranging between 30 and 100 ng/mL [5].

Twenty-four patients underwent a mastectomy combined with immediate reconstruction, whereas 53 patients were treated through breast-conserving surgery (oncoplastic surgery was performed in nine subjects). In particular, after breast surgery, we collected data about the size, type, and histological features of BC.

All the data were analysed by using SPSS 15.0 version for Windows (SPSS, Chicago, IL, USA). For descriptive statistics, the means ± standard deviations (SDs) for parameters with Gaussian distributions (after confirmation with histograms and Kolmogorov–Smirnov tests), and medians and intervals (minimum–maximum) for non-Gaussian variables, were used.

The comparison among normal variables was performed by using one-way ANOVA or Bonferroni tests. We used chi-square (χ2) and Fisher tests for comparisons among the variables of frequency. Pearson linear correlation analysis was used for the calculation of R coefficients. A value of *p* < 0.05 was considered statistically significant. 

The study was conducted according to the guidelines of the Declaration of Helsinki and approved by the Institutional Review Board and Ethics Committee of Fondazione Policlinico Universitario Agostino Gemelli IRCCS, Università Cattolica del Sacro Cuore, Rome, Italy. 

## 3. Results

The principal characteristics of the study participants are summarized in Table 1.

The patients were divided into three main subgroups according to the type of cancer: subgroup one (70 women (mean age: 54.2 ± 12 SD) with ductal carcinoma)); subgroup two (six women (mean age: 54.7 ± 15 SD) with carcinoma in situ); and subgroup three (11 women (mean age: 55.6 ± 14 SD) with lobular carcinoma). 

We did not find significant differences among groups according to the major analysed variables (i.e., the tumour size, expression of oestrogen receptors (ERs), expression of progesterone receptors (PRs), Ki67, human epidermal growth factor receptor 2 (HER2), and median level of 25(OH)D). In particular, the age did not significantly differ among the subgroups of cancers (*p* > 0.05).

A positive significant correlation (R > 0.7) was found between non-invasive carcinoma in situ and 25(OH)D levels and age (R = 0.82) (Figure 1).

A positive but nonsignificant correlation was reported between invasive ductal carcinoma and 25(OH)D and age (R = 0.45). 

A negative but nonsignificant correlation was found between invasive lobular carcinoma and 25(OH)D and age (R = 0.24).

## 4. Discussion and Conclusions

Hypovitaminosis D is notably associated with a loss of bone and muscle strength, and major international guidelines recommend maintaining normal 25(OH)D levels (>20 ng/mL) in vitD-deficient elderly institutionalized patients and postmenopausal women at higher risk of fragility fractures through appropriate supplementation [5,11]. 

Conversely, the benefits of vitD supplementation for skeletal muscle health in community-dwelling, premenopausal women and younger healthy subjects remain questionable [12]. 

According to some evidence [13], hypovitaminosis D could favour the development of several extraskeletal conditions (i.e., autoimmune, inflammatory, cardiovascular, and metabolic diseases), although the effective role of vitD in the aetiology of these pathologies needs to be further elucidated. In particular, the risks of colon, prostate, ovarian, and breast cancer have been associated with vitD deficiency by many authors [14,15].

Abbas et al. [16] found, in a population-based case–control study including 289 premenopausal women and 595 matched controls (aged between 30 and 50 years), a significant inverse association between BC risk and plasma 25(OH)D concentrations (*p* < 0.05). Compared with the lowest category (<30 nmol/L) (OR = 1 (95% CI)), the ORs for higher plasma concentrations of 30–45, 45–60, and ≥ 60 nmol/L were 0.68 (0.43–1.07), 0.59 (0.37–0.94), and 0.45 (0.29–0.70), respectively (*p* for trend = 0.0006). Interestingly, this association was stronger for progesterone receptor-negative BC (PR-), with evidence suggestive of effect heterogeneity (*p* for heterogeneity = 0.05, case-only model). Additionally, the authors found a significantly reduced risk of BC, with an OR of 0.90 (0.84–0.96) per 10 nmol/L increment when considering 25(OH)D as a continuous variable. Meanwhile, no significant interactions between plasma 25(OH)D and the first-degree family history of BC, age at menarche, duration of breastfeeding, parity, alcohol intake, or BMI were reported.

It has been hypothesized that vitD may exert direct protective effects against cancer by promoting cellular apoptosis and differentiation and by inhibiting angiogenesis and tissue inflammation. At the same time, different general risk factors that can favour cancer development by themselves could negatively affect vitD metabolism, such as smoking, obesity, low physical activity, and sun exposure [17,18]. 

Another study by Alipour et al. [19] compared a control group (364 women; mean age: 44.2 years) with a case group (308 women; mean age: 43.2 years; 172 subjects with a benign mass; and 136 subjects with a malignant mass) regarding vitD status. The results of this study show that the median serum 25(OH)D assessed in the case group was lower than that in the control group (7.7 vs. 8.7 ng/mL), and that the median serum 25(OH)D was higher in benign (7.9 ng/mL) than in malignant cases (7 ng/mL). In the comparison between each of these two case groups with controls, the median 25(OH)D was higher in the control group, lower in the group of patients with benign lesions, and the lowest in the group with cancer. However, the differences between the median 25(OH)D in the benign cases and controls, as well as benign cases and cancers, were not statistically significant (*p* = 0.3 and *p* = 0.1, respectively). The histology of four of the 136 BC was in situ ductal carcinoma; the others were invasive ductal carcinomas. In comparison with subjects with euvitaminosis D (25(OH)D > 35 ng/mL), the ORs for BC were 3 (95% CI: 1.11–8.1) in subjects with severe vitD deficiency (25(OH)D < 12.5 ng/mL), 0.96 (95% CI: 0.3- 2.8) in patients with moderate vitD deficiency (25(OH)D between 12.5 and 25 ng/mL), and 1.79 (95% CI: 0.9–3.5) in subjects affected by mild hypovitaminosis D (25(OH)D between 25 and 35 ng/mL), after adjustment of different variables (i.e., age, menarche, parity, menopausal status, breastfeeding, and family history of BC). Thus, for less-severe hypovitaminosis D, the relationship between vitD status and BC risk appeared to be nonsignificant [19].

The main purpose of our retrospective analysis was to find a possible relationship between the biological finding of BC, according to its histological features, and the level of vitD assessed in all the patients before undergoing breast surgery that would have confirmed the suspicious diagnosis of BC. Firstly, we did not find a significant difference among the subgroups with different types of cancer regarding all the analysed variables (age; tumour dimension; expression of ERs, PRs, Ki67, and HER2; and median level of 25(OH)D) (*p* > 0.05). As mentioned above, the available data in the literature highlight the tendency of both benign and malignant breast masses to be associated with lower median levels of 25(OH)D in comparison with those in healthy subjects. However, the difference in terms of vitD status between benign and malignant lesions did not appear to be clearly significant, as also confirmed by our analysis [19]. Additionally, although our study was small, the results seem to be in line with data reporting a decreased frequency of invasive lobular cancers in the last two decades compared with non-invasive in situ and invasive ductal BC [20]. This phenomenon may be explained by the supposed influence of hormonal exposure, which may contribute to facilitating the appearance of invasive lobular BC and may render this type of tumour more susceptible to incidence variation within population studies [21]. Secondly, a positive significant correlation (R > 0.7) was found between non-invasive carcinoma in situ and 25(OH)D and age (R = 0.82), whereas a nonsignificant but negative correlation was found between invasive lobular carcinoma and 25(OH)D and age (R = 0.45). At the same time, a positive but nonsignificant correlation was reported between invasive ductal carcinoma and 25(OH)D and age (R = 0.45).

A previous retrospective case–control study by Peppone et al. [22] on 224 women diagnosed with Stage 0–III BC showed that suboptimal vitD levels (<32 ng/mL) were more common in women with later-stage disease, non-Caucasians, and those who received radiation therapy (*p* < 0.05). More specifically, the ORs for suboptimal vitD levels were 3.15 (95 CI%: 1.05–9.49) for triple-negative vs. non-triple-negative, 2.59 (95 CI%: 1.08–6.23) for ER- vs. ER+, 2.35 (95 CI%: 1.14–4.84) for premenopausal vs. postmenopausal status at diagnosis, and 2.29 (95 CI%: 2.05–4.98) for negative family history vs. positive. On the other hand, the OR for suboptimal vitD levels was 2.22 (95 CI%: 0.86–5.71) for invasive vs. non-invasive BC. Our results seem to be consistent with those of the latter study since our evaluation did not show a clear impact of vitD status on the invasiveness of BC. 

However, our findings show that vitD status and age could be positively correlated for in situ BC, a type of cancer that is generally associated with lower biological aggressiveness and/or invasiveness than the others. Although the evidence regarding the close correlation between vitD deficiency and in situ BC remains limited, these data show a very interesting picture. 

Regarding invasive lobular BC, we observed an inverse, although nonsignificant, correlation between vitD and age, probably due to the small number of patients. These findings may be related, on one hand, to the most frequent biological features of invasive lobular BC, which seem to be influenced more by hormones than by other exogenous and/or endogenous factors, and, on the other hand, to a potentially less-protective role of vitD at the cellular level because of its insufficiency in this type of BC.

According to recent systematic reviews, vitamin D insufficiency is observed in patients with newly diagnosed breast cancer, and supplemental vitamin D intake showed an inverse relationship with this outcome [8,23]. 

A recent secondary analysis of data from the Women’s Health Initiative CaD trial (n = 36,282 cancer-free postmenopausal women aged between 50 and 79 years, randomly assigned to a daily 1000 mg dose of calcium plus 400 IU of vitamin D or to a placebo) found a lower risk of ductal carcinoma in situ (DCIS) throughout approximately 20 years of follow-up (HR = 0.82; 95% CI: 0.70 to 0.96). These results seem to suggest that, since DCIS could be considered a precursor of invasive BC, supplementation with calcium and vitD might reduce BC risk by acting at an early stage in the natural history of the tumour [24]. However, that evaluation has some limitations since it was a post hoc analysis that did not consider calcium and/or vitamin D intake from dietary sources and/or the effects of each supplement separately. 

Interestingly, a multicentre randomized, double-blind, placebo-controlled study conducted in the United States in men ≥ 50 years and women ≥ 55 years without cancer and cardiovascular disease at baseline showed that supplementation with vitamin D3 (cholecalciferol, 2000 IU/d) and marine omega-3 fatty acids (1 g/d) could produce a significant reduction in advanced cancers (metastatic or fatal) compared with placebo (226 of 12,927 assigned to vitamin D (1.7%) and 274 of 12,944 assigned to placebo (2.1%); HR = 0.83 (95% CI: 0.69–0.99); *p* = 0.04), particularly in subjects with normal BMIs (HR = 0.89; 95% CI: 0.68–1.17) [9].

However, the cancer incidence was similar in the treatment and placebo groups. Thus, a clear conclusion about the favourable impact of vitD supplementation on the cancer risk for the general population cannot be drawn [25].

Moreover, there are many known, and still-unknown, endogenous and exogenous mechanisms involved in tumorigenesis; thus, these results need to be critically evaluated [26].

Our study presents several limitations, as it was a retrospective analysis and may have included some selection biases: the relatively small number of participants, the absence of a control group, and the exclusion of other potential confounding factors, such as body mass index (BMI), dietary and lifestyle habits (smoking, alcohol consumption, and sport activities), family history of BC, comorbidities, and medications. 

Taken together, our analysis did not show significant differences among types of BC regarding vitD status. Additionally, we did not find significant differences among subgroups of BC with respect to tumour size and age (*p* > 0.05).

Despite the small number of cases, since we found a positive significant correlation between vitD levels and age for in situ BC, our results seem to suggest a protective role of baseline endogenous vitD levels in in situ BC, different from that in more invasive types of BC. In other words, vitD, through its putative antiproliferative activity at the cellular level [27], could contribute to reducing the invasiveness of cancer cells. 

Therefore, this work provides encouraging data since, even if preliminarily, we can hypothesize that vitD status may affect the occurrence of less-invasive types of BC, rather than others. 

Further prospective multicentre trials with larger numbers of patients and longer follow-up are necessary to draw more validated conclusions. Even if clinical studies investigating the synergistic role of vitD in BC treatment are still inconclusive [28], our results could suggest that ensuring an appropriate level of 25(OH)D could become a promising choice in the field of BC cancer prevention. However, these findings need to be confirmed in larger and well-designed intervention studies.

## Figures and Tables

**Figure 1 jpm-12-00465-f001:**
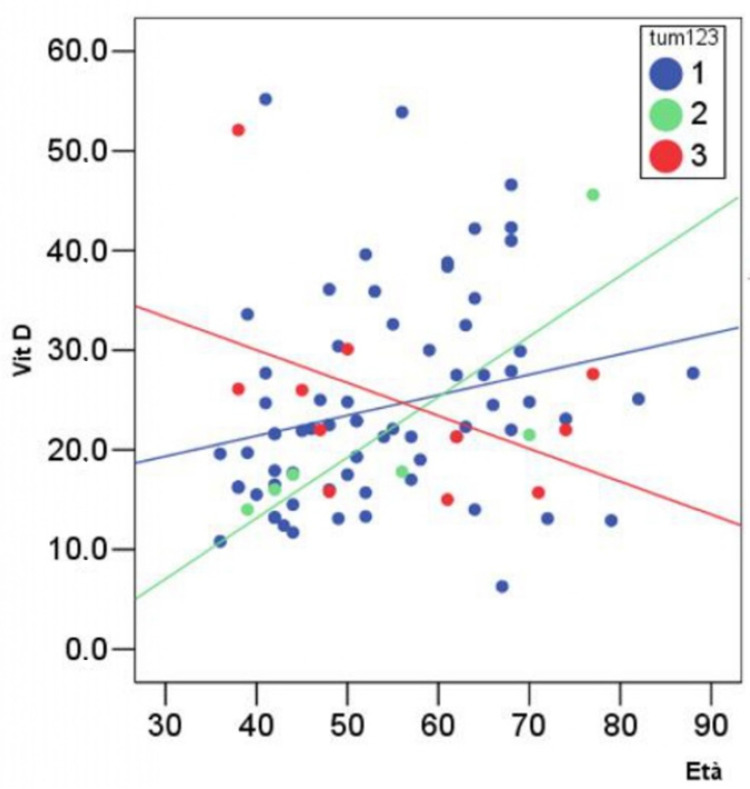
Relationship between age and vitamin D levels in major BC types (tum 1 = ductal BC; tum 2 = in situ BC; and tum 3 = lobular BC).

**Table 1 jpm-12-00465-t001:** Patients’ characteristics and ANOVA correlations between vitamin D levels and tumour type, the type of receptors expressed, and grading.

	Tum	*n*	Mean	SD	Min	Max	ANOVA	Sum of Squares	df	Mean Square	F	Sig.
Age	1	70	54.2	12.0	36	88	Between Groups	16,861	2	8430	0.054	0.948
	2	6	54.7	15.8	39	77	Within Groups	13,218,403	84	157,362		
	3	11	55.6	14.1	38	77	Total	13,235,264	86			
	Total	87	54.4	12.4	36	88						
Vit D	1	70	24.3	10.2	6.3	55.2	Between Groups	33,251	2	16,625	0.155	0.856
	2	6	22.1	11.8	14.0	45.6	Within Groups	8,984,555	84	106,959		
	3	11	24.9	10.4	15.0	52.1	Total	9,017,806	86			
	Total	87	24.2	10.2	6.3	55.2						
yT	1	70	13.6	16.0	0	105	Between Groups	1,034,492	2	517,246	2.094	0.130
(mm)	2	6	14.0	13.0	1	35	Within Groups	20,752,675	84	247,056		
	3	11	24.0	15.3	4	60	Total	21,787,167	86			
	Total	87	14.9	15.9	0	105						
yN	1	66	0.5	0.9	0	3	Between Groups	1184	2	0.592	0.843	0.435
	2	3	0.0	0.0	0	0	Within Groups	51,985	74	0.702		
	3	8	0.3	0.7	0	2	Total	53,169	76			
	Total	77	0.5	0.8	0	3						
ER	1	70	57.8	40.7	0	100	Between Groups	5,156,117	2	2,578,058	1.687	0.191
	2	6	58.5	37.7	1	90	Within Groups	128,387,286	84	1,528,420		
	3	11	81.0	26.8	1	95	Total	133,543,402	86			
	Total	87	60.8	39.4	0	100						
PR	1	70	28.9	37.6	0	98	Between Groups	7,798,377	2	3,899,189	2.677	0.075
	2	6	44.3	48.2	0	90	Within Groups	122,346,542	84	1,456,506		
	3	11	56.1	36.4	1	90	Total	130,144,920	86			
	Total	87	33.4	38.9	0	98						
Ki67	1	70	34.5	25.7	1	80	Between Groups	3,001,866	2	1,500,933	2.552	0.084
	2	1	20.0	.	20	20	Within Groups	46,465,122	79	588,166		
	3	11	17.2	9.7	5	35	Total	49,466,988	81			
	Total	82	32.0	24.7	1	80						
HER2	1	70	1.4	1.1	0	3	Between Groups	5001	2	2500	1.964	0.147
	2	6	1.5	1.4	0	3	Within Groups	106,953	84	1273		
	3	11	0.7	0.9	0	2	Total	111,954	86			
	Total	87	1.4	1.1	0	3						

## Data Availability

The data presented in this study are available on request from the corresponding author.

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
