# Peer review of "Vitamin D and Histological Features of Breast Cancer: Preliminary Data from an Observational Retrospective Italian Study"

_jpm, 2022, doi:10.3390/jpm12030465_

Round 1

Reviewer 1 Report

file attached

Author Response

Dear Editors,

We are pleased to receive your corrections that could make our paper entitled “Vitamin D and Histological Features of Breast Cancer: Preliminary Data from an Observational Retrospective Italian Study” more suitable for publication in “Journal of Personalized Medicine”.

  • Reviewer #1:

Thank you for the suggestions of the Reviewer #1 to improve the quality of our paper. We appreciated your important contribution.

Materials and Methods

  • We added the method used for the measurement of 25OHD in the manuscript (automated Abbott Architect 25OHD immunoassay).
  • ER, PR, HER2 and Ki67 expression was defined according to international guidelines (NCCN 2021) (i.e.: ER+ or PR+ was >1%; Ki67 + was >20%; HER2 + was 3+).
  • According to the suggestions of the reviewer #1, we specified the inclusion and exclusion criteria in the revised manuscript as well as ethical informations about our evaluation. Additionally, more specifically, at the end of the manuscript, there is a specific section including the Institutional Review Board Statement and the Informed Consent Statement.

  • We added in the text the definition of vitamin D deficiency according to the US Endocrine Society Clinical Practice Guidelines (See Holick et al., 2011).

The median level of vitamin D did not differ among groups. Taken together, 19 women had normal levels of 25OHD while 68 women were affected by hypovitaminosis D according to the US Endocrine Society Clinical Practice Guidelines.

Results

  • We corrected the abstract regarding the main purpose of our study in both abstract and introduction. We apologize for the mistakes.
  • Of course, we evaluated all possible significant relationships between vitamin D and other collected parameters (See also Page 3;line 100). However, we chose to highlight only significant results.
  • We did not find a significant correlation between tumor grade and vitamin D levels.
  • We added p-values in the final version of the manuscript
  • We modified the Table 1 reporting only the value of p derived from the ANOVA analysis as suggested by the reviewer #1

Discussion

  • According to the suggestion of the Reviewer #1, we revised the phrase recurring to the adjective “limited” that could be more appropriate. Besides, we added and discuss the reference suggested by the Reviewer #1.

On the other hand, for example, it should be noticed that the Authors of VITAL-HF (Vitamin D and Omega-3 Trial–Heart Failure) found that “recurrent rates of HF hospitalization were not different between vitamin D intervention (341 events) and placebo (364 events; HR, 0.94 [95% CI, 0.81–1.09], P=0.44).  Specifically, in their conclusions, they stated that “interventions with vitamin D or omega-3 fatty acid supplements did not significantly reduce the first HF hospitalization rate”. (See also Supplementation With Vitamin D and Omega-3 Fatty Acids and Incidence of Heart Failure Hospitalization: VITAL-Heart Failure. Circulation 2020). Therefore, in line with our conclusions, the need of further larger interventional studies regarding the role of supplementation of vitamin D in the incidence of cancer and/or other diseases remains advisable. On the other hand, the purpose of our evaluation was not to study the effect of vitamin D supplementation on BC: in fact, it meant to be a retrospective observational analysis rather than an intervention trial.

  • According to the suggestions of the Reviewer #1, we discussed the reference Peila R, Xue X, Cauley JA, Chlebowski R, Manson JE, Nassir R, Saquib N, Shadyab AH, Zhang Z, Wassertheil-Smoller S, Rohan TE. A Randomized Trial of Calcium Plus Vitamin D Supplementation and Risk of Ductal Carcinoma In Situ of the Breast. JNCI Cancer Spectr. 2021 Aug 31;5(4):pkab072. doi: 10.1093/jncics/pkab072. PMID: 34476342; PMCID: PMC8406436.
  • We underlined and critically discussed in different parts of the discussion that the results of the principal available studies need to be interpreted with caution. In fact, sometimes, within the same study that found positive correlation between vitD status and reduced BC risk, a clear significant relationship between vitamin D and BC in subgroup analysis did not emerge (For example, see Peppone et. al; Alipour et al.). On the other hand, our study by itself did not find significant difference regarding vitamin D levels within subgroups thus, in our opinion, our analysis would not appear to be biased.

Conclusions  

  • We changed and better clarified the conclusion of our abstract.
  • We corrected the conclusion hightlighting that “Taken together, our analysis did not report significant differences among subtypes of BC regarding vitD status. Besides, we did not find significant differences among subgroups of BC with respect to tumor size and age (p > 0.05)”. Additionally, we better explained our conclusions.
  • We deleted the phrase suggested by the reviewer #1 such as “Furthermore…ageing

References

  • We revised the reference list. We apologize for the mistakes.

Reviewer 2 Report

This study analyzed retrospective data to investigate the correlation of vitamin D levels with breast cancer, finding that low levels were associated with some types of breast cancer.

The main strength of the study is the contribution of new information about different histological types of breast cancer associated with vitamin D levels.
The main weakness is the lack of plausible biological mechanisms proposed to explain these cancer/vitamin D relationships.

The researchers should not assume that vitamin D supplementation is the solution to the problem of low vitamin D in breast cancer. More studies should be cited investigating controversial findings regarding vitamin D supplementation and cancer. 

The researchers should also cite a wider variety of literature that suggests mediating factors that could cause cancer  while lowering vitamin D, especially dietary factors. For example, dysregulated metabolism of dietary phosphate is associated with cancer, and phosphate intestinal absorption is regulated by bioactive vitamin D (calcitriol) biosynthesized by the kidneys, suggesting that high phosphate levels lower vitamin D levels.

Author Response

Dear Editors,

We are pleased to receive your corrections that could make our paper entitled “Vitamin D and Histological Features of Breast Cancer: Preliminary Data from an Observational Retrospective Italian Study” more suitable for publication in “Journal of Personalized Medicine”.

  • Reviewer #2:

  • We appreciated the punctual observations of the Reviewer #2.

This is a preliminary observational study about the possible impact of vitamin D status on different subtypes of breast cancer. In our opinion, the Reviewer #2 should take into account that the aim of our evaluation was not to study the effect of vitamin D supplementation on BC: in fact, this was not an intervention trial. Conversely, as also indicated in the introduction, the main objective of this evaluation “was to correlate the subtypes of BC with level of vitD”. 

More precisely, we specified in the conclusion that “Taken together, our analysis did not report significant differences among types of BC regarding vitD status. Besides, we did not find significant differences among subgroups of BC with respect to tumor size and age (p > 0.05). However, since we found a positive significant correlation between vitD levels and age for in situ BC, our results seemed to suggest a protective role of baseline endogenous vitD levels in in situ BC, differently from more invasive types of BC. In other words, vitD, through its putative antiproliferative activity at cellular level, could contribute to reduce the invasiveness of cancer cells. Therefore, this work provides encouraging data since, even if preliminarily, we could hypothesize that vitD status may affect the occurrence of less invasive types of BC rather than the others”.

Anyway, according to the interesting suggestions of the Reviewer #2, we enlarged the discussion, furtherly emphasizing that, although the results of some studies seem to suggest that vitamin D could influence the incidence of BC, there are still limited and inconclusive data about vitamin D status and/or supplementation with respect to the prevention of cancer (See Peila R.; Gissel T). On the other hand, certainly, it remains difficult to identify an unquestioned predominance of a single factor in the cancer development. (See also Peters JM, et al).

Anyway, it is precisely because data available in literature are still controversial, that each contribution, although preliminarily, could be useful to increase our knowledge and to encourage further investigations.

Finally, we cited other factors influencing vitamin D status, as suggested by the reviewer #2 (See, in particular, Brown RB. et al)

Round 2

Reviewer 1 Report

The authors have improved the manuscript with the most recent revisions. However, the primary conclusion of the paper—that VitD levels could be associated with less invasive types of BC—is not supported by the data presented. The only statistical significance is the relationship between age of women and VitD status, not between BC subtype and VitD status. Further, their conclusions were based on data where there were only 6 women with in situ disease, which is insufficient to draw strong conclusions. If the authors could increase the sample size of their study, then they could better support their current conclusions (relationship between age and VitD status in women with in situ disease) as well as present their data demonstrating non-significant relationships between vitamin D and other collected parameters, as this data would add to the current body of knowledge.

Author Response

Dear Editors,

We are grateful for your suggestions that could make our paper entitled “Vitamin D and Histological Features of Breast Cancer: Preliminary Data from an Observational Retrospective Italian Study” more suitable for publication in “Journal of Personalized Medicine”.

Reviewer #2

  • We changed and clarified the conclusion of our abstract.

Additionally, according to the suggestions of the reviewer #2, we furtherly specified at different points of the conclusion of our discussion that “Taken together, our analysis did not report significant differences among types of BC regarding vitD status. Besides, we did not find significant differences among subgroups of BC with respect to tumor size and age (p > 0.05). Anyway, although for few cases, since we found a positive significant correlation between vitD levels and age for in situ BC, our results seemed to suggest a protective role of baseline endogenous vitD levels in in situ BC, differently from more invasive types of BC. In other words, vitD through its putative antiproliferative activity at cellular level could contribute to reduce the invasiveness of cancer cells. Therefore, this work may provide encouraging data since, even if preliminarily, we could hypothesize that vitD status may affect the occurrence of less invasive types of BC rather than others”.

Thus, we highlighted that our data are preliminary and that our hypothesis, although intriguing, should be confirmed. On the other hand, the primary objective of our study was different and these findings could be only carefully explained. Anyway, in our opinion, discussing also not significant or apparently weak but interesting data might be useful to promote further investigations in this field.

Sincerely